# Who marries whom and intentions for second child: Using family decision-making power as mediator

Yuan Dang *, Xin Liu

Department of Sociology, Fudan University, Shanghai, China

* doudoubear@outlook.com

## Abstract

Interest in exploring fertility intentions, decisions, or the actual number of children through the perspective of assortative mating has been increasing; however, the mechanisms linking these variables remain unclear. Existing studies have shown that gaps in socio-economic resources between spouses shape intra-household decision-making patterns. Individuals who have the final-say power over homemaking exhibit more bargaining power in family fertility decisions. Based on the 2014 China Family Panel Studies, this research used latent class analysis to obtain the intra-household decision-making variable. A generalized structural equation model was built to examine this potential mediator. The findings reveal that family decision-making power helps to elucidate the relationship between the patterns of assortative mating and fertility intentions. Differences in couples' educational attainment are a key aspect in assessing "who" is in charge of the household. The desire for a second child was greater if husbands had the final say. Participants in marriages where wives held decision-making power reported a lower willingness to have a second child. The mediation effects of "husband-dominated" or "wife-dominated" decision-making were confirmed in hypergamous marriage. Indirect-only mediating effects were found in mid-educated homogamous partnerships and hypogamous marriages. Suppression effects were present in educational homogamy among highly educated individuals.

## Introduction

The worldwide total fertility rate (TFR) has declined drastically, dropping by more than half since the mid-20th century, and is projected to continue declining further [1,2]. This ongoing, accelerating, and seemingly irreversible trend declining birth rates is especially concerning in countries like China where the TFR fell below the replacement rate of 2.1 children per female in the early 1990s and now is down to around 1.1. Governments facing low fertility have introduced various baby-friendly policies to simulate birth rates. Since China implemented the "two-child" and "three-child"

**Data availability statement:** All relevant data are within the paper and its Supporting Information files.

**Funding:** The author(s) received no specific funding for this work.

**Competing interests:** The authors have declared that no competing interests exist.

policies, there has been a proliferation of studies analyzing "who" is willing to have more children or "who" will have more children.

Given that women bear children and the primary responsibility for childcare, research initially focused on indicators leading to heterogeneity in women's reproductive outcomes, suggesting that higher educational attainment and labor force participation by women is one of the primary reasons for delayed childbearing and/or reduced number of children [3]. Furthermore, decisions regarding the use of assisted reproduction technologies (ART) [4] may also influence reproductive outcomes. Other factors interplaying with women's own socio-economic status that influence reproductive decisions, which have often been discussed by researchers in this field, include, but not limited to, age [5], prior fertility [6], *hukou* [7], ethnicity [8], geographic region [9], and household income [10]. However, the traditional (women-only) perspective overlooks the fact that children are the "fruits" of the joint efforts of both spouses. Some scholars [11,12] proposed that the number of children desired is the result of mutual influence and consensus reached between partners; accordingly, a "dyadic-level model" is advocated.

Scholars have increasingly revisited individual fertility intentions, decisions, or actual behaviors from a couple's perspective [13–21]. Hypogamous couples (wives more educated than husbands) have a reduced desire for a "big" family, while willingness to have more children increases when wives have relatively lower education (hypergamy) [14,19–21]. Educational homogamous mating among those highly educated may suppress the ideal or actual number of children [13,18,19]. Accordingly, "who" one marries is associated with different fertility desires or preferences. However, research re-examining fertility from a couple-level perspective is nascent. The mediating mechanisms by which a wife's relative and absolute educational levels facilitate or constrain individual desire for more than one child remain unclear.

Chinese women are outperforming men across the board in tertiary education from undergraduate to postgraduate levels. "Marriage of matching doors (*men dang hu dui*)" has remained a mainstream concept in mate selection [22–24]. The higher a woman's educational level, the greater the expectation for educational achievement of potential spouses. Highly educated females are more inclined to choose husbands with the same level of education. Years of schooling has a delaying effect on the age at first marriage for women [25]. The later a woman enters marriage, the more likely she is to choose a husband whose educational level and income are poorer than her own [26]. Higher absolute and relative educational resources among women increase their likelihood of becoming the primary economic providers in the household [27]. This has reversed the power dynamics of "who" decides family matters, including the desired family size [28].

Lack of consensus between the husband and wife is common regarding the desired number of children [29,30]. Highly educated career women have children later and prefer fewer children [1,31], leading to frequent disagreements with their partners on fertility issues. However, given their significant role in financial support and household decision-making, highly educated women may exert greater influence over family fertility decisions [32]. Phan (2013) [33]emphasizes that increasing

educational levels and labor force participation, along with involvement in household decision-making and contraceptive awareness, are the main components of women's empowerment. Female empowerment has driven the widespread practice of "better birth and better upbringing" in developing countries [34–38].

Traditional Chinese society is bound by patrilineal kinship, along with Confucian ideals of "the husband is the guide of the wife (*qi wei fu gang*)" and "men dominate, women follow (*nan zhunv cong*)" [39]. Women were not allowed to attain education or work outside the home, and their sole mission was to "carry on the family line (*chuan zong jie dai*)" [40]. Education encourages women to break free from domestic confines [41], a crucial prerequisite for extensive participation in wide-ranging socio-economic activities [42,43]. Economic independence has reduced women's reliance on men [44], empowering them to autonomously make decisions related to their personal well-being, including reproductive decisions [33,45–47]. Women's decision-making power helps explain lower fertility levels, longer birth intervals, and lower rates of unintended pregnancies [37].

Although these studies elucidate the relationship between educational assortative mating, "final-say" power in the household, and individual desire for more children, empirical research rarely analyzed whether and how relative educational levels between spouses influence an individual's ideal number of children through household decision-making power. This study aims to explore whether and how the distribution of decision-making power within the household influences individual fertility intentions under different patterns of assortative mating. As illustrated in Fig 1, couples' educational pairing plays an important role in shaping their household decision-making dynamics, where those who hold decision-making power often have more leverage over fertility choices. Family decision-making power functions as a mediator, clarifying the underlying mechanism through which the socio-economic gaps between husbands and wives affect their fertility decisions.

## Literature and research hypotheses

### Conceptualization of household decision-making power

"Power" is a complex and abstract concept, often leading to definitional ambiguities [48]. In the eponymous chapter of their book *Power in Families*, Cromwell and Olson [48]p.5 clarify "family power" as the "potential or actual ability to change the behavior of other family members." They further distinguish three dimensions of power relations within the family system: 1) "Bases of family power," which refers to the resource advantages held by individuals that increase their influence;

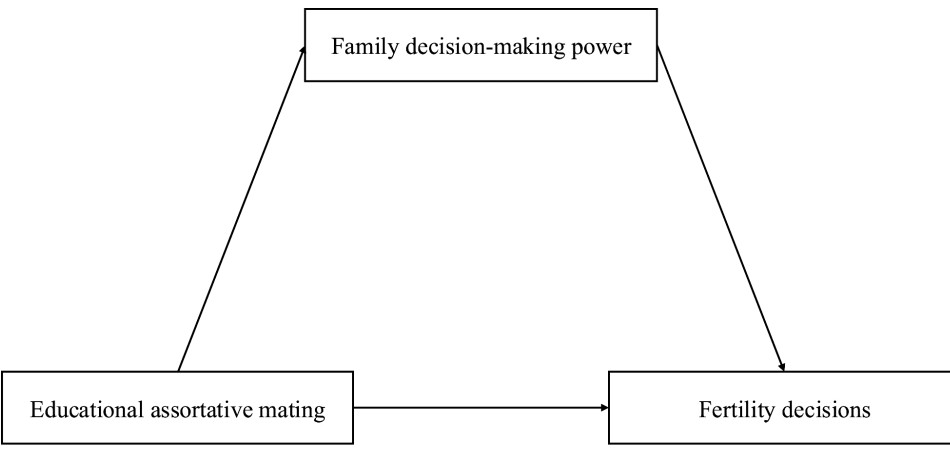

**Fig 1. Mediation analysis illustration.**

2) "Family power processes," indicating the interaction strategies individuals employ to take control in negotiations or decision-making; and 3) "Family power outcomes," i.e., "who" wins the final decision-making power [48]pp.5–7.

This multidimensional analytical framework has been widely adopted [49–52]. Considering the ambiguity of the "process" dimension in household, which makes it difficult to operationalize precisely in quantitative research, studies tend to proceed directly from "bases" to "outcomes" [48,52]. This has led to research primarily focusing on family power to switch to measuring decision-making power in family matters— "whoever" has the say, "whoever" has the power. Therefore, "losing power" implies that one's decision-making is curtailed or exploited.

"Decision-making power in family matters" is not exactly synonymous with "real power in the family." Having the ability to make decisions is one aspect of embodying and understanding "(family) power" [53]. Additionally, overemphasizing and relying solely on "decision-making" limits the scope of our examination of power, significantly neglecting many forms of power in everyday life that are unrelated to decision-making [52]. According to Xu [54], the cost inputs, resource exchanges, and establishment of authority within heterosexual marriages often rely on intangible interactive symbolic exchanges, such as love, mutual respect, housekeeping skills, and service contributions. The strong sense of "family-centered" consciousness in traditional Chinese families has imbued household decision-making power with the characteristic of "caring for the whole family," which may be passively assigned rather than actively acquired [55]p.16. Is the real power in family held by the party that enjoys "the freedom not to decide," or by the one who has to make decisions even if s/he does not want to, temporarily setting aside their freedom (ibid.)?

In this study, decision-making power in family affairs is used as a mediating variable. Given that "household decision-making power" is parsimonious in conveying the actual "family power" in quantitative measures, the term "power" in this study, unless otherwise specified, refers to decision-making authority over household matters.

## Relative socio-economic resources between spouses and who has the "say"

Blood and Wolfe [56]conducted structured interviews with 909 wives in American families and found that husbands' power scores were linked to the resources they possessed (as measured by educational level, income, occupational status, etc.). Their findings provide empirical support for the resource (exchange) theory. Specifically, the relatively more resourceful partner will use the greater economic contributions as "leverage" in exchange for bargaining power over household decisions. The other partner, who is disadvantaged in terms of resources and relies on the financial support of the "power-holder," voluntarily assumes the larger share of domestic chores [57]. Accordingly, women's major share in household labor and deprivation of decision-making power in families is attributable to their economic dependency on their husbands [58].

In middle- and high-income countries, the proportion of women "marrying down" has exceeded that of women "marrying up," driven by rising educational levels and labor market participation rates [59–61]. Improvements in women's educational attainment have contributed to shifting social norms, leading to greater gender equality within the household and redefining women's roles in decision-making processes. Studies [62] have shown that as women gain more education, traditional gender roles are increasingly challenged, facilitating more egalitarian relationships in both family and societal structures. As women progressively hold more "bargaining chips," the balance of "who calls the shots" tilts in favor of women [63,64]. Numerous studies confirm that women's relative or absolute socio-economic status [31,65–69] contributes to their role as family decision-makers. Participation in household decision-making is seen as an important aspect of female empowerment [38,53,70,71]. Some studies [72–76] have also found that husbands increase their share of household labor when both spouses have high educational levels, or when women "marry down." Moreover, greater improvement in women's educational attainment has prompted a shift from hegemonic masculinity to caring masculinity [77]. Therefore, this study hypothesizes that with a rise in women's absolute and relative educational levels, household decision-making power gradually transitions from the traditional "husband calls the shots" to an egalitarian "collaboration" or "wife calls the shots." Specific mating patterns can be further refined into two hypotheses:

H1a: In homogamous marriages, the higher the educational level of both spouses, the lower the likelihood of the husband having decision-making power, and higher the likelihood of joint decision-making or the wife having decision-making power.

H1b: Compared to hypogamous marriages, hypergamy less likely to have joint decision-making or the wife having decision-making power, and more likely that the husband has decision-making power.

**"Motherhood penalty" vs. male "fatherhood reward": Final-say power and desired number of children**

Women, as primary bearers of population reproduction, are constrained by traditional gender roles as "caretakers," shouldering greater childcare responsibilities. The negative impact of childbearing on career prospects, such as wage earnings and promotion opportunities, for women in the workforce is considerable [78–80]. For highly-educated and high-income women, the opportunity cost of having multiple children is particularly high [47,81,82]. However, empowered with decision-making authority over household matters, they possess the confidence to mitigate the cost of childbearing [28]. Essentially, they can make decisions about childbearing according to their subjective agency, even in the face of opposition [47,53]. From this, based on the premise of wives having decision-making authority (including both shared decision-making and exclusive decision-making by the wife), the hypothesis regarding fertility intentions is as follows:

H2a: The greater the likelihood of joint decision-making or the wife having decision-making power, the less likely the desire to have more children.

Men and women face starkly different consequences for childbearing. In contrast to the "motherhood penalty" [83,84], fathers experience a "fatherhood bonus" in terms of wage packages, and higher-income men receiving even greater wage premiums for fatherhood [85–88]. Comparatively, men express a stronger desire for parenthood [89–92]. If women are deprived of final-say power over household matters (including reproductive issues), attributable to their husband's dominance in the household or the influence of "more children, more blessings (*duo zi duo fu*)" concept of fertility culture, individuals find it difficult to autonomously decide on the number of offspring [45]. Doepke and Tertilt [92] suggest that in the climate of male "dictatorship" in household decision-making, the number of children born to a couple is consistent with the male's individual fertility preferences. "Disempowered" women are more likely to desire larger families than women partially involved in family decision-making [93]. Therefore, this study hypothesizes that:

H2b: The greater the likelihood of husbands being in charge of household decisions, the more likely the individual's desire to have more children.

In connection with the aforementioned review of how the intra-couple relative educational achievements affects "who" wins the final-say power over family matters, this study postulates that:

H3: Household decision-making power mediates the relationship between educational assortative mating patterns and fertility intentions.

## Methods

### Data and variables

The data were drawn from the 2014 China Family Panel Studies (CFPS), which employed a multi-stage random sampling to collect data from urban and rural households across 30 provinces (including autonomous regions and municipalities) across China, making the findings broadly generalizable. Despite its longitudinal nature, only the 2014 survey data asked respondents about who had the final-say power in various household matters and the ideal number of children (irrespective of policy constraints). Since out-of-wedlock births are extremely rare in China [94], eligible observations are those who

were married during the survey. Considering that women's abilities to conceive are affected by their biological age, this study refers to Gan and Wang [95], tightening the age range of respondents and their spouses. The lower limit is the legal marriable age in China, 22 for men and 20 for women; the upper age limit for female samples is 49, while for male samples, it is set near retirement age, 59. Overall, 8,238 observations were used for analysis.

The dependent variable is whether there is a willingness to have more than one child. The universal "two-child" policy was not officially implemented until the beginning of 2016. China was under the selective "two-child" period in 2014; couples were permitted to have a second child only if either parent was an only child. The variable in the CFPS questionnaire that collects information on the ideal number of children was deliberately added with the caveat "regardless of policy restrictions." The independent variable is the patterns of educational matching between spouses. CFPS asked respondents and their spouses about their highest level of education, which was categorized into four levels: "primary school or below," "middle school," "high school," and "college or above." Accordingly, "who is higher" and "who is lower" regarding educational attainment within the couple was identified. For homogamous marriages, couples with primary school certificates or below were defined as "low-education homogamy." "Mid-education homogamy" indicates that both spouses completed secondary education (including middle school and high school). The case where both spouses had received higher education degree was termed "high-education homogamy."

The mediating variable analyzed in this study is power distribution in household decision-making. CFPS2014 asked respondents about who ultimately made decisions in terms of "household budget allocation," "savings, investments, and insurance," "real estate purchase," "children's education," and "high-priced consumer goods." This helps determine whether the husband or the wife exercised final-say power over family matters; a value of "0" was assigned to the former and the latter was coded as "1." Previous studies using this variable [96,97] aggregated the scores for each item to generate a variable reflecting women's family power. However, we do not know the weight of each item in constructing this indicator. Summation assumes that each item contributes equally to the final score, ignoring the variabilities between family matters and complexity of family decision-making processes. This study performed LCA on these five manifest variables; three latent classes were obtained. According to the conditional probabilities of each family matter in each class, these three classes were defined as "husband wins final-say power," "joint decision-making," and "wife wins final-say power."

This study also controlled for a number of socio-demographic characteristics: 1) age of the respondent; 2) gender; 3) hukou status (rural hukou coded as "0" and urban as "1"); 4) ethnicity (ethnic minorities coded as "0" and Han Chinese as "1") 5) geographic region (divided into "eastern," "central," "western"; 6) logarithm of logarithm last year's household income; 7) actual number of children. Furthermore, individuals' dependence on the "bread" earned by their spouses [98–101], as well as relative proportion of household labor shared by the individual [102,103], are both likely to influence power distribution in household decision-making. Therefore, wives' dependency on economic resources and wives' relative amount of housework in the family were accounted for in the analytical model. Following previous research [58,104], the formula for calculating wives' economic dependency is: (wife's income – husband's income)/ (wife's income + husband's income). The value ranges from −1 to +1, indicating a gradual decline in wives' economic dependence on husbands. Wives' relative share of housework can be calculated through the formula: wife's housework time/ (wife's housework time + husband's housework time).

## Analytical strategy

This study analyzed the mediating effect of "who" dominates final-say power on how different patterns of educational assortative mating affect individual's intention of more than one child. Prior to GSEM, LCA was used to delve into the essence behind the phenomenon, assuming that whether women hold ultimate decision-making power over various household matters constitutes a multidimensional manifestation of "who" is in charge of household affairs. This part of the statistical analysis was conducted using Mplus 8.0 software.

Initially, the model assumes no association between the observed indicator variables (i.e., "who" ultimately decides those household issues) and starts fitting the latent category model from 1-class, gradually adding the number of latent categories. The optimal number of classes is determined by comparing fit indices such as Akaike's Information Criterion (AIC), Bayesian Information Criterion (BIC), sample-size adjusted BIC, entropy, Vuong-Lo-Mendell-Rubin test (VLMRT), adjusted Lo-Mendell-Rubin test (aLMRT), Bootstrap likelihood-ratio test (BLRT), and class probabilities. The lower AIC, BIC, and aBIC, the better the model fit [105]. Entropy measures the degree of separation among latent classes, with higher values demonstrating clearer distinctions [106]. However, an overfitted model may also show a high entropy value, thus it is not recommended for model selection [106,107]. VLMRT, aLMRT, and BLRT examine whether adding an additional latent class $k$ improves the overall model fit compared to the original $k$-1 latent classes [108]. Additionally, the interpretability of the classes is crucial when considering the best latent class solution. Muthén [109] suggests choosing the more parsimonious $k$-1 class solution if the $k$ class shows only marginal changes in these fit indices and lacks substantive meaning.

In order to estimate and test the mediating effect of household decision-making power, two paths are established in the GSEM model: 1) the effect of intra-couple relative educational achievements on "who" wins decision-making power in the household; 2) the effect of educational assortative mating on ideal family size after controlling for power distribution in household decision-making. To approximate the "net" effect of the mediating variable, the control variables in both paths remain consistent. A major advantage of GSEM is that it allows for the fitting of these two equations within one unified framework, as shown in the following equation:

$$Logit\ P(Decision\_power) = a_{k-1}Education\_match_{k-1} + \gamma_1 Control\_variables + e_1 \tag{1}$$

$$Logit\ P(Fertility\_intention) = c'_{k-1}Education\_match_{k-1} + b_{p-1}Decision\_power + \gamma_2 Control\_variables + e_2 \tag{2}$$

When dealing with an unordered multi-categorical independent variable (*a multi-categorical X*), a widely-used approach is to treat one as the reference group, generating $k$-1 dummy variables [110–112]. The estimated mediation effect values should then be understood as relative indirect effects compared to the reference group [111]. However, studies involving a multi-categorical mediating variable (*a multi-categorical M*) are rare. Referring to Meng et al. [113], a psychiatric study examining multi-categorical mediators, which are the three dimensions (support of family, friends, and significant others) of the social support scale, and three regression models were built to respectively predict these dimensions by the explanatory variable. Similarly, the mediating variables in this study were converted into three dummies: "is the husband decision-maker?" "Do spouses jointly make decisions?" and "is the wife decision-maker?" Three Logit models were applied to obtain coefficient $a$ in Equation (1). Subsequently, three different decision-making patterns were introduced in Equation (2) to obtain coefficient $b$. By examining the significance and effect size of $a \times b$, we can determine the presence of the mediation effect. GSEM was conducted using Stata 18.0 statistical software. The significance levels of the above mediation effects were tested using the bias-corrected non-parametric percentile Bootstrap method (set to resamples 5,000 times and a 95% confidence interval).

## Results

### Descriptive analysis

The descriptive statistics of study variables are presented in Table 1. A majority of respondents (83.14%) intended to have at least two children, and educational homogamy (54.57%) was the dominant preference for mate selection; families where both spouses had a primary school education or below was relatively common (30.21%), whereas families where both spouses had received higher education were relatively rare (6.12%). Among the two types of educational

**Table 1. Descriptive Statistics of the Variables.**

| Variables | Count | Mean (SD)/ % |
|---|---|---|
| **Fertility intentions for a second child** | | |
| No | 1 383 | 16.86% |
| Yes | 6 819 | 83.14% |
| **Patterns of educational mating** | | |
| Hypergamy | 2 444 | 29.80% |
| Low-education homogamy | 2 478 | 30.21% |
| Mid-education homogamy | 1 496 | 18.24% |
| High-education homogamy | 502 | 6.12% |
| Hypogamy | 1 282 | 15.63% |
| **Power in household decision-making** | | |
| Husband-dominated | 4 405 | 53.71% |
| Joint decision-making | 1 573 | 19.18% |
| Wife-dominated | 2 224 | 27.12% |
| **Items to assess decision-making power** | | |
| Household budget allocation | 3 705 | 51.12% |
| Savings, investments, and insurance | 3 677 | 50.74% |
| Real estate purchase | 3 660 | 50.50% |
| Children's education | 3 690 | 50.92% |
| High-priced consumer goods | 3 684 | 50.91% |
| **Age** | 8 202 | 39.57 (7.71) |
| **Gender** | | |
| Female | 4 177 | 50.93% |
| Male | 4 025 | 49.03% |
| ***Hukou* status** | | |
| Rural | 6 156 | 75.05% |
| Urban | 2 046 | 24.95% |
| **Ethnicity** | | |
| Non-Han | 739 | 9.01% |
| Han | 7 463 | 90.99% |
| **Geographical region** | | |
| Eastern | 3 513 | 42.83% |
| Central | 2 430 | 29.63% |
| Western | 2 259 | 27.54% |
| **Log of household income** | 8 202 | −0.01 (0.93) |
| **Actual number of children** | 8 202 | 1.60 (0.83) |
| **Wives' economic dependency** | 8 202 | −0.24 (0.63) |
| **Wives' relative amount of housework** | 8 202 | 0.66 (0.37) |

heterogamy, hypergamous marriages (29.80%) were almost twice as high as hypogamous marriages (15.63%). Regarding decision-making within the household, for each of the five items used to assess this aspect, the proportion of individuals with the power of final-say and those without, was evenly split, each accounting for approximately 50%. For distribution in household decision-making power variable derived from LCA analysis, more than half of the families exhibited husband-dominated decision-making (53.71%). The rest were "wife-dominated" (27.12%) or in a cooperative relationship (19.18%). This suggests the prevalence of the traditional "male-dominated and female-subordinate" marital

relationship across the research sample; this may be explained by the lower proportions of highly educated homogeneous and heterogeneous marriages, where wives are more likely to have the final-say.

Table 1 also shows the mean value of wives' degree of economic dependency, as −0.24. This indicates that their proportion of family income is generally lower, making them economically more dependent on their husbands. The mean value of wives' relative share of housework was 0.66, revealing that wives were the primary bearers of household chores. These two variables may be associated with socio-economic inequalities between couples and therefore necessary to control for.

Table 2 presents the cross-tabulation results of educational matching patterns and decision-making power distribution. Wives' absolute and relative educational levels reflect the power dynamics within the household. For women "marrying up" in education, husbands were the decision-makers in 60.72% of the families, while 37.99% of hypogamous families were "wife-dominated." In homogamous marriages, as the couple's shared educational level increased, the prevalence of "husband-dominated" decision-making decreased, while shared or "wife-dominated" decision-making increased.

## Obtaining latent variables through LCA

Table 3 presents the fitting results of each latent model from 1 to 7 categories. The AIC, BIC, and aBIC decrease with increase in the number of classes, and drop significantly at the 3-class model, and then enter an upward trend from the 4-class model. Although the AIC, BIC, and aBIC indicators of the 4-class model are better than those of the 3-class model, membership probabilities of the latter are distributed in a more balanced way. The 4-class model includes a particularly low percentage of single category, suggesting that increasing an additional class does not improve the model's interpretability. Consequently, the 3-class model was identified as optimal for analysis.

Apart from the probability of respondents belonging to each latent class, the conditional probabilities of each manifest variable for each latent class in LCA results should be emphasized. In Table 4, conditional probabilities represent group

**Table 2. Cross-tabulation of Educational Assortative Mating and Household Decision-making Power.**

| Assortative mating Decision-making power | Hypergamy | Low-edu homogamy | Mid-edu homogamy | High-edu homogamy | Hypogamy | Total |
|---|---|---|---|---|---|---|
| Husband-dominated | 1 484 60.72% | 1 526 61.58% | 687 45.92% | 206 41.04% | 502 39.16% | 4 405 53.71% |
| Joint decision-making | 468 19.15% | 380 15.33% | 305 20.39% | 127 25.30% | 293 22.85% | 1 573 19.18% |
| Wife-dominated | 492 20.13% | 572 23.08% | 504 33.69% | 169 33.67% | 487 37.99% | 2 224 27 12% |
| Total | 2 444 100.00% | 2 478 100.00% | 1 496 100.00% | 502 100.00% | 1 282 100.00% | 8 202 100.00% |

**Table 3. LCA Selection Process for Household Decision-making Power.**

| Classes | AIC | BIC | aBIC | Entropy | VLMRT | aLMRT | BLRT | Class membership probabilities |
|---|---|---|---|---|---|---|---|---|
| 1 | 74064.503 | 74101.150 | 74085.261 | | | | | 100.0 |
| 2 | 49149.626 | 49230.249 | 49195.292 | 0.934 | 0.0000 | 0.0000 | 0.0000 | 36.5/ 63.5 |
| **3** | **47592.832** | **47717.431** | **47663.408** | **0.803** | **0.0000** | **0.0000** | **0.0000** | **29.2/ 52.0/ 18.8** |
| 4 | 47350.083 | 47518.658 | 47445.567 | 0.772 | 0.0000 | 0.0000 | 0.0000 | 28.7/ 5.4/ 15.5/ 50.5 |
| 5 | 47325.462 | 47538.014 | 47445.855 | 0.845 | 0.0002 | 0.0003 | 0.0000 | 29.8/ 1.0/ 11.6/ 7.0/ 50.7 |
| 6 | 47335.778 | 47592.306 | 47481.080 | 0.870 | 0.5069 | 0.5117 | 0.6667 | 5.7/ 51.8/ 3.1/ 1.6/ 8.0/ 29.8 |
| 7 | 47347.778 | 47648.282 | 47517.989 | 0.738 | 0.3386 | 0.3386 | 1.0000 | 67.0/ 4.1/ 5.5/ 1.5/ 29.8/ 50.5/ 1.6 |

**Table 4. Results of LCA.**

| | N | Class member-ship probabilities | Conditional probabilities | | | | |
|---|---|---|---|---|---|---|---|
| | | | Household bud-get allocation | Savings, invest-ments, and insurance | Real estate purchase | Children's education | High-priced con-sumer goods |
| Class 1 – Husband-dominated | 5 856 | 0.520 | 0.001 | 0.002 | 0.000 | 0.172 | 0.034 |
| Class 2 – Joint decision-making | 2 115 | 0.188 | 0.449 | 0.313 | 0.126 | 0.695 | 0.458 |
| Class 3 – Wife-dominated | 3 293 | 0.292 | 0.977 | 0.988 | 0.891 | 0.954 | 0.946 |

heterogeneity and help explain the meanings each latent class specified. Conditional probabilities take values from 0 to 1; the closer it is to 1, the greater probability that the wife wins the final-say power, and the closer it is to 0, the higher the like-lihood that the husband dominates decision-making. In Table 4, each item category in the first classification is approaching 0, indicating the dominance of husband in household decision-making in this group. The third classification is the opposite, with the wife mostly dominating decision-making. In the second classification, the decision-making power for "household budget allocation" and "high-priced consumer goods" distributed equally. Wives were more apt to decide on children's edu-cation, while husbands were more inclined to make financial decisions on "savings, investments, and insurance" and "real estate purchase." Therefore, the second classification implies shared or joint decision-making between spouses.

More than half (52%) of respondents were in households where the husband was the decision-maker. "Joint decision-making" accounts for 18.8%. In 29.2% of households, the wife took the lead on household matters. For the convenience of the subsequent mediation analysis, this latent variable is transformed into three dummy variables: "whether the hus-band is the decision-maker," "whether the couple jointly make decisions," and "whether the wife is the decision-maker."

**Mediation analysis using GSEM.** In Table 5, Models 1, 3, and 5 examine how relative differences in educational achievements between spouses determine who makes decisions. Models 2, 4, and 6 assess how educational assortative mating impacts willingness to have more than one child when the power distribution in household decision-making is controlled for, as well as the effect of "who" wins final-say power on fertility expectations is also examined after controlling for mating pattern. Low-education homogamy is set as the reference category.

Model 1 shows a significant decreasing trend in the likelihood that husbands held decision-making power over family matters as the wife's relative educational level increased. The husband's dominance in decision-making was most pro-nounced in cases of women "marrying up," whereas hypogamous matching significantly weakened husbands' authority. Correspondingly, Model 5 indicates that heterogeneous marriages with "wives high and husbands low" were the most favorable for wives having the final say. Among the three levels of educational homogamy, compared to low-educational homogamy, the likelihood for husbands to dominate decision-making was marginally lower when both spouses have middle-level education than higher education (Model 1). However, from the wife's perspective, high-education homog-amy did not significantly predict whether the wife would win the final-say power (Model 5). In Model 3, higher relative and absolute educational level of the wife contributed to couples' "joint decision-making." In high-education homogamy, the likelihood of spouses making joint decisions was marginally higher than in hypogamous marriages. Therefore, Hypothesis H1a is partially supported, and Hypothesis H1b is confirmed.

Husbands being "in charge" had a positive effect on individuals' willingness to have more children (Model 2), and Hypothesis H2b is thus confirmed. When wives held decision-making power, the desire for a second child was suppressed (Model 6). The "joint decision-making" pattern did not significantly impact fertility intentions (Model 4), partially validating Hypothesis H2a. In these three models, the negative impact of women "marrying down" on the desire for more children is not significant. Only the coefficients for women "marrying up" and high-education homogamy are significant, and both are positive. Compared to the reference group defined in this study—the low-educational homogamy—both hypergamy and high-education homogamy increased individuals' motivation to have a second child. The desire for more children was strongest when both spouses are highly educated.

**Table 5. Path Coefficients of the GSEM Model.**

| | Model 1 | Model 2 | Model 3 | Model 4 | Model 5 | Model 6 |
|---|---|---|---|---|---|---|
| | Husband-dominated | Fertility intention | Jointly decided | Fertility intention | Wife-dominated | Fertility intention |
| | Coefficients (SE) | Coefficients (SE) | Coefficients (SE) | Coefficients (SE) | Coefficients (SE) | Coefficients (SE) |
| Hypergamy | 0.031** (0.014) | 0.023** (0.011) | 0.023* (0.012) | 0.024** (0.011) | −0.054*** (0.013) | 0.022** (0.011) |
| Low-education homogamy | *Reference* | *Reference* | *Reference* | *Reference* | *Reference* | *Reference* |
| Mid-education homogamy | −0.092*** (0.017) | 0.002 (0.013) | 0.030** (0.014) | −0.000 (0.013) | 0.062*** (0.015) | 0.002 (0.013) |
| High-education homogamy | −0.086** (0.027) | 0.087*** (0.020) | 0.058** (0.022) | 0.085*** (0.020) | 0.029 (0.025) | 0.086*** (0.020) |
| Hypogamy | −0.165*** (0.018) | −0.015 (0.013) | 0.055*** (0.014) | −0.018 (0.013) | 0.109*** (0.016) | −0.015 (0.013) |
| Husband-dominated | | 0.021** (0.008) | | | | |
| Joint decision-making | | | | 0.006 (0.010) | | |
| Wife-dominated | | | | | | −0.030** (0.010) |
| Control variables | *Controlled* | *Controlled* | *Controlled* | *Controlled* | *Controlled* | *Controlled* |
| N | 8 321 | 8 202 | 8 321 | 8 202 | 8 321 | 8 202 |
| Log Likelihood | −9023.202 | | −7266.047 | | −8164.093 | |
| AIC | 18112.4 | | 14598.09 | | 16394.19 | |
| BIC | 18344.28 | | 14829.97 | | 16626.06 | |

As for the control variables, the coefficients can be found in Table S1 of the appendix. Overall, the responses of those who had more than one child were likely to differ from those who did not. The actual number of children not only significantly affected the respondents' intentions for a second child but also had a significant effect on "who" holds decision-making power within the household. The significance of the other control variables varied across different family decision-making patterns, suggesting independence among the three types of such patterns identified in this study. Additionally, regarding the intentions to have a second child, there were no statistically significant differences across socio-demographic groups (e.g., age, gender, household income, ethnicity). Although age was included as a control variable in the analysis, with the average age of the sample being 37 years, the coefficient for age was not significant in the model. This suggests that within the study, age did not have a significant impact on fertility intentions. While advanced age may influence fertility intentions, especially in terms of increased risks associated with later childbearing, this study did not find a direct effect of age on fertility intentions.

Table 6 reports the estimated relative indirect effects of "who" owns decision-making power mediating the effects of educational mating patterns on fertility intentions. The indirect effect is the product of coefficient *b* (the effect of "who" is in charge on desired family size) and coefficient *a* (the effect of educational assortative mating on power distribution in decision-making). If the confidence interval does not contain zero, the effect is significant. According to Table 6, overall, "who" holds decision-making power mediates the relationship between educational matching patterns and fertility intentions, validating the mediation Hypothesis H3. Since the coefficient for "joint decision-making" in Model 4 is not significant, meaning coefficient b is not significant, referring to the mediation mechanism testing process pointed out by Wen and Liu [114], the Bootstrap method is recommended for testing the significance of *a* × *b*. This study has found that for all four types of educational assortative mating (as reference to the low-education homogamy), the indirect effects of the "joint decision-making" show insignificance; thus these effects are omitted in Table 6.

**Table 6. The Mediation Effect-sizes and Confidence Intervals of a Multi-categorical M for a Multi-categorical X.**

| | Husband-dominated | Total effects | Wife-dominated | Total effects |
|---|---|---|---|---|
| Hypergamy | **0.0006**<br>**[0.0001, 0.0017]** | **0.0240**<br>**[0.0050, 0.0442]** | **0.0016**<br>**[0.0006, 0.0032]** | **0.0240**<br>**[0.0049, 0.0441]** |
| Low-education homogamy | *Reference* | *Reference* | *Reference* | *Reference* |
| Mid-education homogamy | **−0.0019**<br>**[-0.0038, -0.0005]** | −0.0003<br>[-0.0253, 0.0246] | **−0.0019**<br>**[-0.0038, -0.0007]** | −0.0002<br>[-0.0252, 0.0247] |
| High-education homogamy | **−0.0018**<br>**[-0.0043, -0.0005]** | **0.0849**<br>**[0.0414, 0.1257]** | −0.0009<br>[-0.0031, 0.0004] | **0.0849**<br>**[0.0414, 0.1255]** |
| Hypogamy | **−0.0034**<br>**[-0.0064, -0.0010]** | −0.0180<br>[-0.0454, 0.0085] | **−0.0033**<br>**[-0.0060, -0.0013]** | −0.0179<br>[-0.0453, 0.0085] |

Although husbands' dominance in household decision-making positively predicts individuals' desire for a second child, women's relative educational attainments weakened the possibility of men as decision-makers. Table 6 indicates that, in the case of women "marrying up," the husband-dominated power distribution facilitated a positive effect of this mating pattern on the desire for more children. In Model 2, the coefficient for high-education homogamous marriage is significantly positive, while the indirect effect of "husband-dominated" decision-making on the relationship between this mating pattern and motivations toward having a second child is significantly negative (Table 6). This represents a typical suppressing effect [115,116], at which point we should calculate the size of the suppressing effect by taking the absolute value of the ratio of the indirect effect to the direct effect [117]. Husband's "disempowerment" offset 2.3% ($|\frac{-0.002}{0.087}|$)of the positive impact of high-education homogamy on fertility intentions.

In Model 2, the direct effect *c'* of mid-education homogamous marriage as well as of hypogamy are not statistically significant. Some scholars [112,118–120] argue that this indicates a case of full mediation, implying that the mechanism by which these two types of assortative mating affect the willingness to have more than one child depends on whether the husband wins the final-say power. Conversely, other scholars [112,114,121] pointed out that the concept of "full mediation" may hinder further exploration of other potential mediating or suppressing variables. Since the total effects are not significant for these two mating patterns, calculating the proportion of the indirect effect to the total effect does not exhibit statistical significance. Wen & Liu [114]suggest that if the proportion of the indirect effect exceeds 90%, using the term "full mediation" may be acceptable. Therefore, this study adopts the concept of "indirect-only mediation" [122] as an alternative to "full mediation."

According to Models 5 and 6, a higher relative educational level of the wife contributes to her dominance in household decision-making, and the likelihood of wanting a second child becomes negative. The direct effect of mid-education homogamous marriage or hypogamy on fertility intentions is not significant in Model 6. However, Table 6 shows that the indirect effects of whether wives dominated household power-making are significant in the aforementioned relationships, indicating the existence of only an indirect effect. Hypergamous marriage posed challenges to the wife-dominated decision-making power structure, leading to stronger fertility intention. The indirect effect of wives' decision-making power, therefore, is positive, and its effect size (0.0016) is somewhat stronger than that of husbands' decision-making power (0.0006). The mediator shows no significant indirect effect in the relationship between high-education homogamy and desire for more than one child.

This study further performs sensitivity tests for female and male samples separately, as detailed in Tables S3 and S5. Table S2 presents the descriptive statistics stratified by gender. Consistent with the above results for the full sample, women's relative and absolute educational levels are related to the decision-making power structure in their household, and "who" wins the final-say power helps explain the heterogeneity in childbearing desires predicted by educational assortative mating. However, the significance of "who" has the final-say power varies between men and women (see Tables S4 and S6). For males, there are no significant indirect effects of "husband-dominated" and "joint decision-making"

for all four types of educational matching (compared to low-educational homogamous marriage). Nonetheless, the mediating effect of wife-dominated decision-making power is positive and significant for hypergamy. The lower possibilities of winning decision-making power over household matters owing to women's relatively lower educational levels related to men's desire to have more children. In line with the full sample, indirect-only mediation effects present in mid-educational homogamy and hypogamy for male respondents.

For female respondents, no significant indirect effect of "joint decision-making" is found. The direct effect of high education homogamous marriage on fertility intention is positive, while the indirect effect of husband-dominated power structure is negative, revealing a suppressing effect, that is, 2.7% ($\left|\frac{-0.002}{0.074}\right|$)of the positive direct effect is suppressed by the husband as power-holder in household decision-making. Significant indirect-only mediation effects of "husband-dominated" decision-making, without statistically proven significant direct effects are observed for both mid-education homogamy and hypogamy. Additionally, "wife-dominated" power structure does not significantly mediate the relationship between high-educational homogamous marriages and fertility intentions. Only indirect effects are presented for three types of educational pairing.

## Discussion

Since the implementation of the "three-child" policy, questions regarding whether it can effectively address the persistent decline in (re)fertility intentions have been raised. Thus far, the trend of continuously declining fertility rates has not been reversed. The relatively high costs of raising children, combined with women's struggle to balance career and family, significantly impact fertility intentions in China [123,124]. Furthermore, economic downturns that occurred during the pandemic exacerbated financial burdens, leading to a decreased willingness among women to have more children [125]. As women gain greater access to education and employment opportunities, they are more likely to assert control over household decision-making, including reproductive choices. Their growing financial independence allows them to have a stronger voice in shaping family planning decisions, thereby influencing the intention to have children [126–129]. In line with previous research, this study further confirms that the increased economic and social status of women relative to their husbands plays a central role in influencing family dynamics and fertility intentions, signifying a shift in decision-making power.

This study has certain limitations that must be addressed. We cannot arbitrarily attribute ultra-low fertility intentions to improvement of women's autonomy and bargaining power in both the household and society. The decision-making power over the five family issues in the CFPS2014 reflects not only "who" is more influential in the household, but also the capability of "running the household." Housekeeping ability is assumed to be associated with the party with "gendered advantage" [130]. As the home sphere is stereotyped to be headed by women, managing the household naturally is a woman's responsibility. The one who holds household decision-making power "worries (*cao xin*)" for the sake of serving family members [55] p.16. It may be paradoxical to see that the wife, who is in charge of household finances management, is also the sole performer of domestic labor [103]. In this study, descriptive statistics (Table 1) also revealed that wives continued to be primary bearers of household chores. Therefore, could the decision-making power in household become a burden aggravating the challenges a career woman faces when balancing unpaid domestic labor and paid employment (i.e., the double burden), which in turn, attribute to the decline in fertility intentions?

To verify this supposition, dual causality is involved, and single year cross-sectional data is insufficient to clarify the chronological ordering of the relationship. As mentioned earlier, household decision-making may be the "outcome" of gendered division of roles in household [102,103]. Conversely, it is also possible that the magnitude of power is the "cause" of the proportion of household chores one contributes to [131]. The one who is "in charge" of the household is more likely to successfully persuade the other to become involved in household chores [132]. These puzzles require further in-depth research in the future.

Another limitation is its focus on analyzing the mediating effect of decision-making power on fertility intentions from a socio-economic (i.e., educational stratification) perspective. This was to explore the subtle link between educational

assortative mating and fertility intentions. However, this approach overlooks the role of social and cultural factors, such as gender roles/expectations, personal beliefs about child-rearing, and how these norms influence the relationship between household decision-making power and fertility intentions. Family decision-making patterns, as well as the impact of decision-making power on fertility intentions may be influenced by the intersection of both socio-economic and cultural factors. Future research should adopt an interactive perspective to further clarify how different spousal educational pairings shape family fertility intentions and decisions.

Additionally, an individual's contribution to socio-economic resources within a family is not constant. Rather, the distribution of decision-making power is a dynamic process that can change over time, leading to short- and long-term differences in how such power affects fertility intentions. In the short term, socio-economic disparities between spouses may directly influence fertility decisions. However, over time, the redistribution of socio-economic status and resources may result in changes in decision-making power, which can impact long-term fertility intentions, such as whether to delay or forgo the next birth cycle. Therefore, future research should consider the dynamic nature of power distribution in household decision-making, as well as the shifts in who holds the final say within the family at different points in time. This would provide deeper insights into the mechanisms behind the heterogeneity of fertility intentions across various social groups.

### Implications of the findings

The findings of this study have important implications for social and family policy development. First, this research shows that decisions regarding whether to have children, how many to have, and when to have them, are the result of mutual influence and joint decisions made by spouses. Therefore, policy development and implementation aimed at creating a fertility-friendly social system should place the family, rather than the individual, at the core. Accordingly, policies could focus on providing targeted support to families, such as offering parental leave that is equally accessible to both partners; creating more affordable and high-quality childcare services; and providing tax benefits or subsidies for families with multiple children. By supporting family dynamics and considering the shared responsibilities of both spouses in fertility decisions, policies can better address the needs and challenges faced by families, fostering a more supportive environment for childbearing.

Second, policies should consider the barriers faced by women with high socio-economic status, who may experience conflicting pressures between career advancement and family life. The results suggest that when women hold decision-making power, fertility intentions tend to be lower. To address these barriers, policies could offer targeted support for such women, such as flexible work arrangements, enhanced work-life balance initiatives, and career development opportunities that also allow for family planning. By reducing these pressures, policies could potentially help mitigate the negative impact on fertility intentions among women in higher socio-economic brackets.

### Conclusion

Examining the dynamic interaction between couples, particularly through the lens of socio-economic gaps, offers a more comprehensive and nuanced understanding of fertility intentions and real-life practices as fertility decisions are a family matter shaped by the mutual influence of both partners. This study has confirmed that different patterns of socio-economic resource disparities between spouses—referred to as assortative mating patterns—are key variables in demonstrating fertility intentions. It further demonstrates that the distribution of decision-making power within marital relationships mediates the association between who marries whom in terms of education and the individual's fertility intentions of having more than one child. This study affirmed that the relative disparity in educational attainments between spouses is critical for measuring "who" holds decision-making power over household matters. In homogamous marriages where both spouses had lower education levels, or in heterogamous marriages where the wife had a relatively lower degree than her husband, the husband dominated in decision-making power structure.

Both the absolute and relative educational levels of wives contribute to a "jointly decided" or "wife-dominated" family power structure. This study did not indicate the presence of gendered display in power distribution [133], specifically, "doing gender [136]" under the traditional gender ideology of "men's power over women" even when the wife had superior socio-economic status. Instead, the "who" in "whoever" wins final-say power has a gender-neutral character. Educational attainment helps women exert paramount influence (including "household budget allocation," "savings, investments, and insurance," "real estate purchase," and "high-priced consumer goods") over household finances as well as child rearing decision.

Individuals tend to make decisions favorable to them. Women may prefer fewer children owing to the "motherhood penalty," while men may desire larger families driven by "fatherhood premium." This study has found that both spouses' fertility intentions were subordinate to the preferences of the "power-holders" (i.e., "who" was in charge of household matters), highlighting the differences in fertility intentions between "power-holding" women in decision-making and women who have experienced "disempowerment" in households. Sensitivity tests conducted on sub-samples by gender revealed that men's fertility intentions decreased as the likelihood of their wives "in charge" increased. Concurrently, women's desire for a second child became stronger when husbands held household decision-making power, reflecting a reciprocal influence between the spouses.

Since the key assumption of this study was built on the integration of a multi-categorical independent variable (i.e., educational pairing) and a multi-categorical mediator (i.e., power distribution in decision-making), the indirect effects of the latter varied for different mating patterns. In hypergamy, male dominance in household decision-making facilitated positive attitudes toward having a second child, and simultaneously, women winning the final-say power also promoted stronger fertility intentions. For mid-education homogamy or women "marrying down," final-say power distribution exhibited a mediation-only effect, with non-significant direct effects of mating patterns on fertility intentions. When both spouses had higher education, husband-dominated decision-making structure showed a suppressing effect—a specific case of mediation effect—serving as an inhibitor on the positive direct effect.

## Supporting information

**S1 Table. Path Coefficients of the GSEM Model (Full).**
(DOCX)

**S2 Table. Descriptive Statistics of the Variables (By Gender).**
(DOCX)

**S3 Table. Path Coefficients of the GSEM Model (Male Samples).**
(DOCX)

**S4 Table. Mediation Effects (Male Samples).**
(DOCX)

**S5 Table. Path Coefficients of the GSEM Model (Female Samples).**
(DOCX)

**S6 Table. Mediation Effects (Female Samples).**
(DOCX)

## Acknowledgments

We would like to thank the editors and reviewers for their time and helpful suggestions. We would also like to thank all the listeners at the regular Tuesday night's seminar for their comments on an earlier draft.

## Author contributions

**Conceptualization:** Yuan Dang.

**Formal analysis:** Yuan Dang.

**Methodology:** Yuan Dang.

**Software:** Yuan Dang.

**Supervision:** Xin Liu.

**Validation:** Yuan Dang.

**Visualization:** Yuan Dang.

**Writing – original draft:** Yuan Dang.

**Writing – review & editing:** Yuan Dang.

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
