## [Decision Letter · Decision Letter 0]

Dear Dr. Dang,

Thank you for submitting your manuscript to PLOS ONE. After careful consideration, we feel that it has merit but does not fully meet PLOS ONE’s publication criteria as it currently stands. Therefore, we invite you to submit a revised version of the manuscript that addresses the points raised during the review process.

Please pay particular attention to queries raised on the conceptualisation and methodology, especially considering other factors that may influence fertility decision.

We look forward to receiving your revised manuscript.

Kind regards,

Abiodun Adanikin, Ph.D

Academic Editor

PLOS ONE

Journal Requirements:

Reviewers' comments:

Reviewer's Responses to Questions

**Comments to the Author**

1. Is the manuscript technically sound, and do the data support the conclusions?

Reviewer #1: Yes

Reviewer #2: Partly

Reviewer #3: Yes

2. Has the statistical analysis been performed appropriately and rigorously?

Reviewer #1: Yes

Reviewer #2: I Don't Know

Reviewer #3: Yes

3. Have the authors made all data underlying the findings in their manuscript fully available?

Reviewer #1: Yes

Reviewer #2: Yes

Reviewer #3: Yes

4. Is the manuscript presented in an intelligible fashion and written in standard English?

Reviewer #1: Yes

Reviewer #2: Yes

Reviewer #3: No

Reviewer #1: 1. The abstract must end with a strong conclusive statement based on the key results of the study. This affirmative conclusion that unites the results is absent.

2. The topic and keywords must show the geographical context of the study.

3. The introduction must present a global perspective to the discourse on family fertility decisions before presenting the China context. Recent debates on the use of Artificial Reproductive Technology for family fertility decisions and assortative mating decision-making power must be discussed to enrich the literature while bringing it up to speed with current literature.

4. The literature review section is interesting and offers enough theoretical context for the study and it's hypotheses set.

5. The methods section is great and would highly ensure replication of the study's results.

6. The results have been scholarly presented, likewise the conclusion and discussion sections.

7. In the conclusion, the study's key limitations must be exemplified.

Reviewer #2: The manuscript describes how fertility decisions are influenced by assortative mating through a decision making mediator variable.

However, there are several flaws in the conceptualization and presentation that make it difficult to follow.

1. Introduction: There is a robust literature on fertility decisions and factors influencing these decisions but this was not discussed in the background. Instead there was a long discussion about marriage choices. The introduction could be tightened and a conceptual framework included could help to tie together the analysis. In addition, the purpose of the paper is unclear. In one place the author states "This paper aims to establish a generalized structural equation model (GSEM) to estimate the indirect effect of “who” is in charge in family affairs on individual fertility intentions under different patterns of assortative mating. The bias corrected percentile bootstrap method was used for significance testing. The mediating variable defined in this study—intra-household decision-making power—is not an existing variable in the dataset, but obtained through latent class analysis (LCA)" This statement is more about the methods than how a typical study purpose is written and later there are six separate hypotheses. Highlighting one primary study question in specific terms would help to clarify the paper.

2. Methods: The methods section is not written in a standard format and mixes results into the methods with table 3.1 Table 3.1 should be in the results and should also be stratified by gender and the individual items for LCA should be included so that we can see the distributions by sex before analysis.

3. The absence of a description of the factors that influence fertility decisions in the introduction presents a problem in the analysis because there is no acknowledgement about how respondent age, prior fertility, and other factors may influence the outcomes. These are important variables as respondents who have a child already are likely to respond differently than those who do not. Age is also a factor as the median age of the sample is 37. Higher risks may emerge for women bearing children after the age of 35 and this too could be considered in this analysis.

Reviewer #3: Peer Reviewer Comments

My Comments:

1. Abstract: Your abstract is quite comprehensive and informative. Here are a few suggestions to improve clarity and readability:

Abstract:

Interest in exploring fertility intentions, decisions, and the actual number of children through the perspective of assortative mating has been increasing; however, the mechanisms linking these variables remain unclear. Existing studies have shown that gaps in socio-economic resources between spouses shape intra-household decision-making patterns. Individuals who have the final-say power over homemaking exhibit more bargaining power in family fertility decisions. Based on the 2014 China Family Panel Studies, this research used latent class analysis to obtain the intra-household decision-making variable. A generalized structural equation model was built to examine this potential mediator.

The findings reveal that differences in couples' educational attainment are a key aspect in assessing "who" is in charge of the household. The desire for a second child was greater if husbands had the final say. Participants in marriages where wives held the decision-making power reported a lower willingness to have a second child. The mediation effects of "husband-dominated" or "wife-dominated" decision-making were confirmed in hypergamous marriages. Indirect-only mediating effects were found in mid-educated homogamous partnerships and hypogamous marriages. Suppression effects were present in educational homogamy among highly educated individuals.

2. Reduce the keywords to the following suggested ones:

Keywords: Educational assortative mating, fertility intention, household decision-making power, latent class analysis, generalized structural equation modeling

3. How do educational and socio-economic differences between spouses influence household decision-making power and fertility intentions?

o Exploring this question can help to clarify the mechanisms through which assortative mating impacts family dynamics and fertility decisions.

4. What role do cultural and societal norms play in shaping decision-making power within households and fertility intentions?

o Understanding the influence of cultural contexts can provide a more comprehensive view of how these factors affect family planning.

5. How does the balance of decision-making power within households affect the intention to have a second child in different demographic groups (e.g., urban vs. rural, different age groups)?

o Investigating these variations can offer insights into how different populations experience and respond to these dynamics.

6. What are the short-term and long-term impacts of decision-making power dynamics on fertility intentions and outcomes?

o This question can help to identify patterns and trends over time, offering a deeper understanding of the causal relationships.

7. How do existing family policies and economic conditions influence decision-making power and fertility intentions?

o Examining the external factors that affect household dynamics can provide valuable information for policymakers.

8. To what extent do gender roles and expectations influence the mediation effects of decision-making power on fertility intentions?

o Investigating this can shed light on the gender-specific aspects of decision-making and how they impact family planning.

9. How do personal beliefs and values regarding family size and child-rearing influence the decision-making power and fertility intentions of spouses?

o Understanding the personal dimensions can add depth to the analysis of household dynamics.

10. What are the implications of your findings for social and family policy development?

o Considering the practical applications of your research can enhance its relevance and impact.

11. Separate the Discussion section from the conclusion section.

12. What are the strengths and limitations of this study?

13. After the Discussion section, can you include a sub-heading on “Implications of the findings”

14. Ask for a Professional English editor to address all grammatical errors and lexis structure of your manuscript.

15. Kindly check your references and see if they are fully citied in the main content of your research. Kindly check and revisit them all.

By addressing these questions, you can delve deeper into the complex relationships between marital patterns, decision-making power, and fertility intentions, ultimately enhancing the robustness and relevance of your research.

**Do you want your identity to be public for this peer review?** For information about this choice, including consent withdrawal, please see our Privacy Policy

Reviewer #1: **Yes: ** Dickson Adom

Reviewer #2: No

Reviewer #3: **Yes: ** Dr Monica Ewomazino Akokuwebe

---

## [Author Response · Author response to Decision Letter 1]

30 Mar 2025

Response to the reviewer’s comments

Reviewer #1:

1. The abstract must end with a strong conclusive statement based on the key results of the study. This affirmative conclusion that unites the results is absent.

Response:

For the abstract, I have added a strong conclusive statement at the beginning of the second paragraph. Please refer to page 2.

2. The topic and keywords must show the geographical context of the study.

Response:

I have added a new keyword “fertility in China” to that list. Please refer to page 2.

3. 1) The introduction must present a global perspective to the discourse on family fertility decisions before presenting the China context. 2) Recent debates on the use of Artificial Reproductive Technology for family fertility decisions and assortative mating decision-making power must be discussed to enrich the literature while bringing it up to speed with current literature.

Response:

1) A global perspective has been incorporated at the beginning of the Introduction section, please refer to page 3.

2) The use of ART is now mentioned, I have added a reference supporting the point “higher educational attainment and labor force participation by women is one of the primary reasons for the decisions regarding the use of assisted reproduction technologies (ART),” as seen on page 3. However, since this is not the primary focus of this study, I have not elaborated further on this point, but rather included it to align with current literature, as per your suggestion.

4. The literature review section is interesting and offers enough theoretical context for the study and its hypotheses set.

Response:

I thank the reviewers for their thoughtful insights.!

5. The methods section is great and would highly ensure replication of the study's results.

Response:

Thank you very much.

6. The results have been scholarly presented, likewise the conclusion and discussion sections.

Response:

Thank you very much.

7. In the conclusion, the study's key limitations must be exemplified.

Response:

I have outlined several limitations of this study, which can be found in the Discussion section on pages 35-38. As suggested by Reviewer#3, I have separated the Discussion section from the Conclusion section.

Reviewer #2:

1. Introduction: 1) There is a robust literature on fertility decisions and factors influencing these decisions but this was not discussed in the background. Instead there was a long discussion about marriage choices. The introduction could be tightened and a conceptual framework included could help to tie together the analysis. 2) In addition, the purpose of the paper is unclear. In one place the author states "This paper aims to establish a generalized structural equation model (GSEM) to estimate the indirect effect of “who” is in charge in family affairs on individual fertility intentions under different patterns of assortative mating. The bias corrected percentile bootstrap method was used for significance testing. The mediating variable defined in this study—intra-household decision-making power—is not an existing variable in the dataset, but obtained through latent class analysis (LCA)" This statement is more about the methods than how a typical study purpose is written and later there are six separate hypotheses. Highlighting one primary study question in specific terms would help to clarify the paper.

Response:

1) “Other factors” (that interplay with women’s own socio-economic status to influence their reproductive decisions) are now included in the Introduction, with supporting references added. Please note that I have only listed in bold the factors included in the models as control variables. Please refer to pages 3-4.

Additionally, I have now included a figure to illustrate the mediating process in the conceptual framework, which I believe makes it clearer and more concise. Please refer to pages 6-7.

2) The main purpose of this study has been highlighted at the end of the Introduction section. You can find this on page 6.

2. Methods: The methods section is not written in a standard format and mixes results into the methods with table 3.1 Table 3.1 should be in the results and should also be stratified by gender and the individual items for LCA should be included so that we can see the distributions by sex before analysis.

Response:

I have moved Table 3.1 to the Results section, and the individual terms for LCA has been included in this table. Please check pages 19-21. Table 3.1 has been stratified by gender, and the corresponding data can be found in the Appendix – S2 Table. Since the gender segregation serves as supporting evidence, I have opted not to include it in the main text to keep the presentation more succinct.

3. The absence of a description of the factors that influence fertility decisions in the introduction presents a problem in the analysis because there is no acknowledgement about how respondent age, prior fertility, and other factors may influence the outcomes. These are important variables as respondents who have a child already are likely to respond differently than those who do not. Age is also a factor as the median age of the sample is 37. Higher risks may emerge for women bearing children after the age of 35 and this too could be considered in this analysis.

Response:

As responded above, brief descriptions of the factors are now included in the Introduction. Additionally, a caution regarding the interpretations of the age-related patterns has been noted. Please refer to page 30 in the Results section. The coefficients of age variable in all 3 models are not significant, indicating that age does not appear to have a substantial impact on fertility intentions within this sample.

Reviewer #3:

1. Abstract: Your abstract is quite comprehensive and informative. Here are a few suggestions to improve clarity and readability:

Abstract:

Interest in exploring fertility intentions, decisions, and the actual number of children through the perspective of assortative mating has been increasing; however, the mechanisms linking these variables remain unclear. Existing studies have shown that gaps in socio-economic resources between spouses shape intra-household decision-making patterns. Individuals who have the final-say power over homemaking exhibit more bargaining power in family fertility decisions. Based on the 2014 China Family Panel Studies, this research used latent class analysis to obtain the intra-household decision-making variable. A generalized structural equation model was built to examine this potential mediator.

The findings reveal that differences in couples' educational attainment are a key aspect in assessing "who" is in charge of the household. The desire for a second child was greater if husbands had the final say. Participants in marriages where wives held the decision-making power reported a lower willingness to have a second child. The mediation effects of "husband-dominated" or "wife-dominated" decision-making were confirmed in hypergamous marriages. Indirect-only mediating effects were found in mid-educated homogamous partnerships and hypogamous marriages. Suppression effects were present in educational homogamy among highly educated individuals.

Response:

I thank the reviewers for their thoughtful suggestions and insights, which have enriched the manuscript and produced a better and more balanced account of the research.

I have revised the abstract in line with your suggestions. Please refer to page 2 for the updated version.

2. Reduce the keywords to the following suggested ones:

Keywords: Educational assortative mating, fertility intention, household decision-making power, latent class analysis, generalized structural equation modeling

I have updated the keywords list based on your suggestions. Please refer to page 3 for the revised version.

3. How do educational and socio-economic differences between spouses influence household decision-making power and fertility intentions?

Exploring this question can help to clarify the mechanisms through which assortative mating impacts family dynamics and fertility decisions.

Response:

I have included a figure illustrating this mechanism into the main text; please see pages 6-7. I hope the relationships between these three variables are now clearer.

4. What role do cultural and societal norms play in shaping decision-making power within households and fertility intentions?

Understanding the influence of cultural contexts can provide a more comprehensive view of how these factors affect family planning.

Response:

I have added “Improvements in women’s educational attainment have contributed to shifting social norms, leading to greater gender equality within the household and a redefined role for women in decision-making processes,”, with supporting reference, in the Introduction section. Additionally, this study focuses on the socio-economic perspective to examine the differences in fertility intentions across various groups. Future studies could explore how this intersects with socio-cultural perspective, which could be considered a limitation of the current study. Please refer to pages 37-38 in the Discussion section.

5. How does the balance of decision-making power within households affect the intention to have a second child in different demographic groups (e.g., urban vs. rural, different age groups)?

Investigating these variations can offer insights into how different populations experience and respond to these dynamics.

Response:

I controlled for various demographic groups in the models. A concise version of the results table is provided in the main text, where the results of all control variables are not listed, but a “controlled” label indicates that these variables were accounted for during the analysis. I have added the table in the Appendix as supporting information; please refer to S1 Table.

6. What are the short-term and long-term impacts of decision-making power dynamics on fertility intentions and outcomes?

This question can help to identify patterns and trends over time, offering a deeper understanding of the causal relationships.

Response:

This could be considered a limitation of the study. An individual’s contribution to socio-economic resources within the family is not fixed. Rather the distribution of decision-making power itself is a dynamic process that can change over time. This fluidity leads to short-term and long-term differences in how decision-making power affects fertility intentions. I have included this point as one of the limitations. Please refer to page 37.

7. How do existing family policies and economic conditions influence decision-making power and fertility intentions?

Examining the external factors that affect household dynamics can provide valuable information for policymakers.

Response:

I have highlighted this point in the beginning sentences of the Discussion section. I noted out that the implementation of the “three-child” policy has not yet reversed the ongoing declining fertility rates. Additionally, economic downturns resulting from the pandemic have exacerbated financial burdens, leading to decreased willingness among women to have more children. As women gain greater access to education and employment opportunities, they are more likely to assert control over household decision-making, including reproductive choices. Please refer to page 35.

8. To what extent do gender roles and expectations influence the mediation effects of decision-making power on fertility intentions?

Investigating this can shed light on the gender-specific aspects of decision-making and how they impact family planning.

Response:

As noted in point 4 above, this study primarily focuses on the socio-economic perspective to examine differences in fertility intentions across various groups. However, future research could explore how this socio-economic perspective intersects with the socio-cultural perspective, which could be considered a limitation of the current study. Please refer to pages 37-38 in the Discussion section for further details.

9. How do personal beliefs and values regarding family size and child-rearing influence the decision-making power and fertility intentions of spouses?

Understanding the personal dimensions can add depth to the analysis of household dynamics.

Response:

As noted in point 4 above, this study primarily focuses on the socio-economic perspective to examine differences in fertility intentions across various groups. However, future research could explore how this socio-economic perspective intersects with the socio-cultural perspective, which could be considered a limitation of the current study. Please refer to pages 37-38 in the Discussion section for further details.

10. What are the implications of your findings for social and family policy development?

Considering the practical applications of your research can enhance its relevance and impact.

Response:

I have added two important implications in the Discussion section.

11. Separate the Discussion section from the conclusion section.

Response:

The Discussion section has now been separated from the Conclusion section for better clarity.

12. What are the strengths and limitations of this study?

Response:

Please refer to the revised Conclusion and Discussion sections, where the strengths and limitations of this study are now discussed in detail.

13. After the Discussion section, can you include a sub-heading on “Implications of the findings”

Response:

I have included “Implications of the findings,” as a sub-section.

14. Ask for a Professional English editor to address all grammatical errors and lexis structure of your manuscript.

Response:

I have submitted the revised manuscript to the T&F editing service.

15. Kindly check your references and see if they are fully citied in the main content of your research. Kindly check and revisit them all.

Response:

I have updated all the references I cited.

---

## [Decision Letter · Decision Letter 1]

Dear Dr. Dang,

Thank you for submitting your manuscript to PLOS ONE. After careful consideration, we feel that it has merit but does not fully meet PLOS ONE’s publication criteria as it currently stands. Therefore, we invite you to submit a revised version of the manuscript that addresses the points raised during the review process.

We look forward to receiving your revised manuscript.

Kind regards,

Abiodun Adanikin, Ph.D

Academic Editor

PLOS ONE

Journal Requirements:

Additional Editor Comments:

**Please revise the conclusion as suggested by reviewer 1** . Thank you.

Reviewers' comments:

Reviewer's Responses to Questions

**Comments to the Author**

Reviewer #1: (No Response)

Reviewer #3: All comments have been addressed

2. Is the manuscript technically sound, and do the data support the conclusions?

Reviewer #1: Yes

Reviewer #3: Yes

3. Has the statistical analysis been performed appropriately and rigorously?

Reviewer #1: Yes

Reviewer #3: Yes

4. Have the authors made all data underlying the findings in their manuscript fully available?

Reviewer #1: Yes

Reviewer #3: Yes

5. Is the manuscript presented in an intelligible fashion and written in standard English?

Reviewer #1: Yes

Reviewer #3: Yes

Reviewer #1: Thanks for revising the manuscript based on my earlier comments. However, the discussion strands where you have meticulously cited previous authors in the concluding section have to be removed. Ensure that the concluding section pivots solely on inferences made from the key results that marries well with the stated study's objectives. Thanks

Reviewer #3: The Authors have addressed all comments. The study is an impactful one and i recommend the manuscript for possible publication.

**Do you want your identity to be public for this peer review?** For information about this choice, including consent withdrawal, please see our Privacy Policy

Reviewer #1: **Yes: ** Dickson Adom

Reviewer #3: **Yes: ** Monica Ewomazino Akokuwebe, PhD

---

## [Author Response · Author response to Decision Letter 2]

13 May 2025

Reviewer #1:

Thanks for revising the manuscript based on my earlier comments. However, the discussion strands where you have meticulously cited previous authors in the concluding section have to be removed. Ensure that the concluding section pivots solely on inferences made from the key results that marries well with the stated study's objectives. Thanks!

Response:

Thank you for your suggestions! All citations to previous authors have been removed from the conclusion.

Reviewer #3:

The Authors have addressed all comments. The study is an impactful one and I recommend the manuscript for possible publication.

Response:

Thank you very much.

---

## [Editor Report · Decision Letter 2]

Who marries whom and intentions for second child: using family decision-making power as mediator

PONE-D-24-24868R2

Dear Dr. Yuan Dang,

We’re pleased to inform you that your manuscript has been judged scientifically suitable for publication and will be formally accepted for publication once it meets all outstanding technical requirements.

Kind regards,

Abiodun Adanikin, Ph.D

Academic Editor

PLOS ONE
---

## [Editor Report · Acceptance letter]

PONE-D-24-24868R2

PLOS ONE

Dear Dr. Dang,

I'm pleased to inform you that your manuscript has been deemed suitable for publication in PLOS ONE. Congratulations! Your manuscript is now being handed over to our production team.

Kind regards,

on behalf of

Dr. Abiodun Adanikin

Academic Editor

PLOS ONE